# Cardiovascular Stress and Characteristics of Cold-Induced Vasodilation in Women and Men during Cold-Water Immersion: A Randomized Control Study

**DOI:** 10.3390/biology11071054

**Published:** 2022-07-13

**Authors:** Lydia Tsoutsoubi, Leonidas G. Ioannou, Konstantinos Mantzios, Styliani Ziaka, Lars Nybo, Andreas D. Flouris

**Affiliations:** 1FAME Laboratory, Department of Physical Education and Sport Science, University of Thessaly, 42100 Trikala, Greece; lydiatsoutsoubi@gmail.com (L.T.); ioannoulg@gmail.com (L.G.I.); konstantinosmantzios@gmail.com (K.M.); stelaziaka@gmail.com (S.Z.); 2Department of Nutrition, Exercise and Sports, August Krogh Building, University of Copenhagen, 2100 Copenhagen, Denmark; nybo@nexs.ku.dk

**Keywords:** CIVD, sex, gender, water immersion, heart rate, sweat rate, core temperature, skin temperature, mean arterial pressure, pain

## Abstract

**Simple Summary:**

Cold-induced vasodilation is a phenomenon that refers to a paradoxical increase in finger temperature that sometimes occurs during cold exposure. Differences between sexes in cold-induced vasodilation have been explored in only a handful of studies. These studies investigated finger skin temperature but did not evaluate toe skin temperature, blood flow in the fingers or toes, clothing, as well as potential underlying mechanisms of cutaneous vasomotion. On the whole, our knowledge on the potential impacts of sex differences on CIVD is limited and this may have important implications for workers and how they cope with exposure to cold environments. The aim of the study was to investigate and compare cold-induced vasodilation and other cardiovascular responses between genders, during exposure to different environmental conditions. The present study demonstrated that women experienced elevated cardiovascular strain and higher frequency of CIVD reactions, particularly in the toes, compared to their male counterparts during cold-water immersion.

**Abstract:**

Background: Cold-induced vasodilation (CIVD) is a phenomenon that refers to a paradoxical increase in finger temperature that sometimes occurs during cold exposure. The aim of this study was to compare CIVD responses between women and men, during exposure to different environmental conditions. Methods: Seven men and seven women participated in a matched controlled study consisting of a familiarization protocol followed by three experimental sessions (cool (10.8 °C WBGT), thermoneutral (17.2 °C WBGT), and hot (27.2 °C WBGT)). In each session, participants were asked to immerse their left hand and foot in warm water (35 ± 1 °C) for five minutes. Thereafter, the left hand and foot were immersed in cold water (8 ± 1 °C) for 40 min. After that, the left hand and foot were removed from the water and participants remained seated for five minutes. Results: For a matched thermal stress, women experienced an elevated cardiovascular strain (heart rate and in some cases mean arterial pressure) and higher frequency of CIVD reactions (men: 31 vs. women: 60) in comparison to their male counterparts. Conclusions: The present study demonstrated that women experienced elevated cardiovascular strain and higher frequency of CIVD reactions, particularly in the toes, compared to their male counterparts during cold-water immersion.

## 1. Introduction

Cold-induced vasodilation (CIVD) is a phenomenon that has been investigated since 1930 [1,2,3,4,5,6,7] and refers to a paradoxical increase in finger and toe temperature that sometimes occurs during cold exposure [1,2,8,9,10]. For this reason, it has been hypothesized that CIVD may act as a protective mechanism to reduce the risk of cold injury in the limbs [5,11,12,13]. In this light, some authors support the notion that CIVD reactions are driven by central mechanisms caused by an increase in deep body temperature [3,4,14,15] while others suggest that it may emerge as a response to peripheral stimuli in which arteriovenous anastomoses play a key role [8,16,17]. Despite the plethora of studies conducted on CIVD, the triggering mechanism of this phenomenon remains unclear, and its appearance has been described as nonsystematic and highly heterogenous [12,18]. Indeed, CIVD is reportedly affected by many factors which may lead to an unpredictable occurrence of the phenomenon. For instance, intra-individual differences such as acclimatization or adaptation to cold [19], alcohol consumption [20], diet [21], exposure to hypoxic environments [22], menstrual cycle, mental stress [23], physical fitness, and smoking [24], as well as interindividual differences such as age [25], sex [10,26,27,28], ethnicity [29], and long-term adaptation to cold have been previously included in the long list of potential factors involved in CIVD [1,8,10]. The majority of these factors have been repeatedly investigated uncovering the importance of inter- and intra-individual variability for the occurrence and magnitude of CIVD [1,8,10], but there is still limited understanding on this matter.

Differences between sexes on the CIVD response have been explored in only a handful of studies [10,26,27,28]. These studies investigated finger skin temperature but did not evaluate, toe skin temperature, blood flow in the fingers or toes, important contributing factors such as clothing, as well as potential underlying mechanisms of cutaneous vasomotion. On the whole, our knowledge on the potential impacts of sex differences on CIVD is limited and this may have important implications for workers and how they cope with exposure to cold environments. This knowledge gap is important to address because most modern workplace practices involve females in the same work tasks as men, and female workers can be found doing jobs in the cold that were previously predominantly occupied by men, such as butchering [30], fishing [31], and performing army-related tasks [32,33].

The aim of the present study was to investigate and compare CIVD and other cardiovascular responses between women and men, during exposure to different environmental conditions, while evaluating an array of physiological and perceptual parameters that may have a contributing role in the occurrence of the CIVD phenomenon.

## 2. Materials and Methods

### 2.1. Ethical Approval and Participants

Participants were requested to provide a medical history, used to exclude patients with Raynaud’s syndrome/phenomenon and those under prescription medication for hypertension or other drugs that could affect vasomotion. The minimum required sample size for investigating “repeated measures, within-between factors” was calculated using the results of a previous repeated-measures study (18) which assessed CIVD reactions in men who were in three different conditions, more specifically in thermoneutral, mildly hypothermic, and mildly hyperthermic conditions. Specifically, we calculated the effect size (f) [34] for the comparisons among fingers and among participants, for each condition, reported in the previously published study regarding minimum and maximum finger temperature (T_f_) [13]. To ensure high statistical power, our calculation of the minimum required sample size was based on the lowest effect size identified. This was an effect size (f) equal to 0.5 (equating to an effect size (d) of 1.0) for the comparison of maximum T_f_ in the mild hypothermic (8.3 ± 1.7 °C) versus the thermoneutral (11 ± 3.2 °C) condition [13]. Sample size calculations were conducted using G*Power 3.1.9.4 [35], while setting statistical power and α error probabilities at 0.90 and 0.05, respectively. Based on this calculation, 6 participants per group (12 participants in total) would provide enough power to detect differences between men and women in our study. Accordingly, seven male and seven female subjects matched for age, body mass index, and body surface area (i.e., *p* > 0.05 between groups) participated in the study. The anthropometric characteristics of all 14 participants are shown in Table 1.

The experimental protocol (ClinicalTrials.gov ID: NCT04215939) conformed to the standards set by the Declaration of Helsinki and was approved by the Bioethics Review Board of the University of Thessaly Department of Physical Education and Sport Science (protocol no.: 1320). Four out of fourteen participants (two men and two women) were smokers and were asked to refrain from smoking at least 10 h prior to each experimental session. The menstrual cycle was not controlled since there is recent evidence demonstrating that women’s heat dissipation [36] and finger blood flow [37] do not differ between menstrual phases. All volunteers were given a full explanation of all the procedures and signed a written informed consent prior to participating in the study.

### 2.2. Experimental Protocol

The study included a familiarization protocol followed by three experimental sessions. During the familiarization session, participants were informed about all data collection procedures/equipment and underwent anthropometric and body composition (dual-energy X-ray absorptiometry) assessments. Each of the three experimental sessions was performed inside a 32.5 m^3^ environmental chamber (HGX22G/190-4, Bock GmbH, Frickenhausen,,Germany) under different conditions: cool (wet-bulb globe temperature: 10.8 °C; air temperature: 16 ± 1°C; relative humidity: 45 ± 5%; air velocity: 0.2 m/s; solar radiation: 0 W/m^2^), thermoneutral (wet-bulb globe temperature: 17.2 °C; air temperature: 23 ± 1 °C; relative humidity: 45 ± 5%; air velocity: 0.2 m/s; solar radiation: 0 W/m^2^), and hot (wet-bulb globe temperature: 27.2 °C; air temperature: 34 ± 1 °C; 45 ± 5% relative humidity; air velocity: 0.2 m/s; solar radiation: 0 W/m^2^) environments. The overall thermal stress experienced by the participants in our study was expressed by means of wet-bulb globe temperature which was found to be the most efficacious thermal-stress indicator for assessing the physiological strain [38,39,40]. The present study is a randomized control trial. The experimental sessions were administered in a random order, based on a random allocation algorithm implemented in Excel Spreadsheets (Microsoft Office, Microsoft, Washington, DC, USA).

For each session, participants arrived at the lab at the same time of the day. They had been asked to refrain from caffeine for at least two hours, from food for at least three hours, and from alcohol and exercise for at least 12 h. Upon arrival, participants wore standard clothing consisting of a long-sleeve (100% cotton) shirt and a pair of pants (100% cotton) to avoid the influence of different fabric properties on physiological data in each experimental session (Figure 1). The room temperature in the lab where the participants changed clothes before entering the environmental chamber was thermoneutral (wet-bulb globe temperature: 16.3 °C; air temperature: 22 °C; relative humidity: 45%; air velocity: 0.2 m/s; solar radiation: 0 W/m^2^). The estimated clothing insulation was 0.67 clo for men (underwear: 0.04; long-sleeve shirt: 0.29; pants: 0.34) and 0.69 clo for women (underwear: 0.04; bra: 0.02; long-sleeve shirt: 0.29; pants: 0.34) [41,42]. Thereafter, participants entered the environmental chamber (with the trial order [temperature] counterbalance, but randomly allocated to each participant) and assumed a seated position for 10–15 min to complete instrumentation. Once all sensors were placed, participants were requested to relax seated for 20 min with their hands supported at the level of the heart (Figure 1). After the baseline period, participants were asked to simultaneously immerse their left hand (up to the ulnar styloid process) and left foot (up to the lateral malleolus) in warm water (35 ± 1 °C) for five minutes to ensure similar starting local tissue temperature [2]. Thereafter, the left hand and foot were immersed in cold water (8 ± 1 °C) for 40 min (Figure 1 and Figure 2). After the end of the cold-water immersion period, the left hand and foot were removed from the water and participants remained seated for an additional five minutes to monitor the recovery phase. The total duration of the experiment was 70 min: 20 min baseline period, five minutes warm-water immersion, 40 min cold-water immersion, and five minutes recovery.

The water tanks used for the warm immersion were 0.58 m^3^ for the hand and 0.85 m^3^ for the foot. The water tanks used for the cold immersion were 0.91 m^3^. Water temperature was controlled within 1 °C using an air-to-water heat pump (model KF120-B, Dongguan, China) and the water was continuously circulating to ensure uniform exposure. Throughout the protocol, both hands were positioned at the level of the heart. During the water immersion periods, the left foot was placed in a water tank on the floor, and the right foot rested on the floor (Figure 1). Water-permeable mesh fabrics were used to line the bottom of the water tanks and the floor to ensure that the immersed hand and foot made no contact with the bottom of the water tanks, and that the non-immersed foot made no contact with the floor of the environmental chamber (Figure 1).

### 2.3. Measurements

Gastrointestinal (T_gi_) and weighted mean skin (T_sk_) temperatures (both assessed in 1 min intervals), as well as finger temperature (T_f_), toe temperature (T_t_), skin blood flow (SkBF), sweat rate, and heart rate (HR) were continuously monitored. Raw data from these variables—assessed in 0.05 s (for SkBF), 1 s (for HR), 8 s (for T_f_ and T_t_), and 10 s (for sweat rate) intervals—were used to provide 1 min averages. In addition, 20 s averages were calculated for T_f_, T_t_, and SkBF and were used to identify CIVD reactions. In terms of the remaining measurements, arterial blood pressure was assessed via automatic auscultation at the non-immersed hand every 10 min during baseline, at the end of the warm-water immersion, every 10 min during cold-water immersion, and at the end of the recovery period. Thermal comfort and thermal sensation were recorded every 10 min during baseline and every five minutes during cold-water immersion. Pain sensation was assessed at the start of cold-water immersion and then every five minutes. Tactile sensitivity of the immersed limbs was tested at the end of the baseline period, at the end of the warm-water immersion, as well as at minutes 5 and 40 of the cold-water immersion (for the latter two, the hand and foot were removed from the water for ~30 s).

The T_gi_ was monitored using telemetric capsules (BodyCap, Caen, France) as an indicator of central body core temperature. Although the time following ingestion does not significantly influence the validity of telemetry capsule measurements of core temperature in the absence of fluid consumption [43], the capsules were always ingested at the same time before the start of the protocol for each participant. Skin temperature from four sites (arm, chest, leg, and thigh) was recorded using wireless thermistors (iButtons type DS1921H, Maxim/Dallas Semiconductor Corp., Sunnyvale, CA, USA) and was used to calculate T_sk_ according to Ramanathan [T_sk_ = 0.3 × (chest + arm) + 0.2 × (thigh + leg)] [44].

Using surgical tape (3M Transpore Tape, 3M Canada, Inc., London, ON, Canada), six ceramic chip skin thermistors (MA-100, Thermometrics, Northridge, CA, USA) were attached on the lower part of the pad on the 2nd finger (i.e., index) of both hands and on the lower part of the pad on the 1st, 3rd, and 5th toe of the immersed foot, as well as on the 1st toe of the non-immersed foot. Data were recorded using a data logger (Smartreader 8 Plus, ACR, Vancouver, BC, Canada).

The SkBF was monitored with a laser Doppler flowmeter (PF4000 LDPM, Perimed, Stockholm, Sweden and PF5010 LDPM, Perimed, Stockholm, Sweden) at the pad of the 2nd finger in each hand as well as at the distal edge of the 1st toe of each foot. The probe (PR 407 small straight probe, Perimed) in the non-immersed 2nd finger was held in place with a plastic mini holder (diameter: 5 mm; PH 07-5, Perimed), which was fixed to the skin using double-sided adhesive strips (PF 105-3, Perimed) without constricting the finger. All other probes (413 Integrating Probe, Perimed, Stockholm, Sweden) were held in place with a plastic holder (PH 13, Perimed, Stockholm, Sweden). The SkBF data were expressed in perfusion units (PU).

Sweat rate was measured at the forehead and at the belly of the gastrocnemius muscle using a ventilated capsule system that utilizes a digital mass flow meter sensor (SFM4100, Sensirion, Staefa, Switzerland). Heart rate was monitored using a wireless heart rate system (Polar Team2, Polar Electro Oy, Kempele, Finland). Arterial blood pressure was assessed at the non-immersed hand via automatic auscultation (Omron Healthcare, M6 comfort, Kyoto, Japan). Mean arterial pressure was calculated as follows [45]:MAP=DBP+(SBP−DBP)/3
where, MAP is the mean arterial blood pressure (mmHg), DBP is the diastolic blood pressure (mmHg) and SBP is the systolic blood pressure (mmHg).

In addition, thermal comfort (1 = comfortable to 5 = extremely uncomfortable) and thermal sensation (−3 = cold to +3 = hot) were recorded using standard scales [46]. Pain intensity (1 = no pain to 10 = worst pain imaginable), as well as pain distress (in Likert (1 = no pain to 10 = unbearable pain) and visual analog scale (left edge = no pain; right edge = unbearable pain)) were recorded using standardized scales [2]. Tactile sensitivity of the tip of the middle finger and the second toe of the immersed limbs was assessed using Semmes–Weinstein monofilaments and a digital esthesiometer, based on previous methods [2,47]. For both measurements, lower values indicate increased tactile sensitivity.

### 2.4. Statistical Analysis

The 20 s average values for T_f_ and T_t_ were used to identify CIVD reactions based on previous criteria (Figure 2) [2,8] as follows:Number of waves (n): the minimum temperature increase to define a CIVD response was set at 1 °C.Minimum temperature (T_min_): the lowest temperature just before the onset of CIVD in °C.Maximum temperature (T_max_): the highest temperature during the CIVD in °C.Onset time (T_onset_): the time from immersion to T_min_ in minutes.Peak time (T_peak_): the time to reach the maximum temperature (time at T_max_ minus time at T_min_) in minutes.Average temperature (T_avg_): the average temperature during the cold-water immersion period without the first five minutes of the cold-water immersion in °C.Temperature amplitude (ΔT): the difference between T_min_ and T_max_ in °C.

Cardiovascular strain (systolic, diastolic, and mean arterial blood pressure, as well as HR) during the first minute of cold-water immersion was calculated by means of delta (Δ) values expressed as the difference between the last minute of the warm-water immersion and the 1st minute of the cold-water immersion.

Chi-square tests were used to compare the frequency of CIVD reactions across (1) the three different environments, (2) the fingers, (3) the toes, as well as (4) the limbs (i.e., the immersed hand vs. the foot). Paired sample t-tests along with effect sizes (d) were used to examine potential differences in the physiological strain (T_f_, T_t_, SkBF, T_sk_, T_gi_, arm/chest/leg/thigh skin temperature, perceptual scales (thermal comfort, thermal sensation, pain), systolic blood pressure, diastolic blood pressure, and HR) experienced between women and men. The comparisons were performed separately for each phase of the protocol (baseline, warm immersion, cold immersion, and recovery). Additional paired-sample t-tests with effect sizes (d) were conducted to examine potential differences in the cardiovascular strain during the first minute of cold-water immersion between women and men. The level of statistical significance in the t-tests was adjusted for multiple comparisons using the Bonferroni correction, resulting in an adjusted level alpha value of 0.008. The magnitude of Cohen’s (d) effect sizes was determined as follows: d (0.01) = very small; d (0.2) = small; d (0.5) = medium; d (0.8) = large; d (1.2) = very large; and d (2.0) = huge [48]. Effect sizes (d) were computed with Excel Spreadsheets (Microsoft Office, Microsoft, Washington, DC, USA) and all other statistical analyses were conducted with SPSS v27.0 (IBM, Armonk, NY, USA).

## 3. Results

Under identical experimental conditions that led to a similar heat strain between women and men (i.e., no clinically meaningful differences in T_sk_, T_gi_, or sweat rate; see subsection “other physiological responses”), our study demonstrated that the studied women experienced increased cardiovascular strain and more frequent CIVD reactions, especially during exposure to the cool environment. Detailed information is presented in the following subsections.

### 3.1. Frequency of CIVD

The total number of CIVD reactions in fingers and toes was 91 (men: 31, women: 60) across all three environments: 14 in the cool, 22 in the thermoneutral, and 55 in the hot environment (*p* < 0.001; Figure 3). Men showed no CIVD in the foot during the cool and neutral exposures. Chi-square analyses revealed that women had more CIVD reactions in the foot in comparison to their male counterparts (*p* < 0.001). No statistically significant differences between sexes in CIVD occurrence were detected in the hand or in both limbs together across all environments (Figure 3).

For CIVD reactions detected in the toes, two participants experienced CIVDs in the cool trial (men: zero, women: two; Table 2), one in the thermoneutral (men: zero, women: one; Table 3), and ten in the hot (men: four, women: six; Table 4) environment. When considering all participants simultaneously, chi-square analysis showed that a significantly higher number of participants demonstrated CIVD reactions in the toes during exposure in the hot environment (n = 10) compared to the thermoneutral environment (n = 1; *p* < 0.001) and the cool environment (n = 2; *p* < 0.001). Overall, we found significant differences in the number of participants demonstrating CIVDs in the toes across groups and environments (i.e., participants with CIVD in the toes × group × environment; *p* < 0.001), with CIVDs being more prevalent for women in the heat compared to the thermoneutral and cool environments. Men demonstrated no CIVD reactions in the toes in cool and thermoneutral environments.

For the CIVDs detected in the fingers, four women revealed CIVD reactions in the cool environment, five in the thermoneutral environment, and five in the hot environment, revealing no statistically significant differences across the three environments (*p* = 0.458). In the male group, five participants revealed CIVDs in the fingers in the cool environment, three in the thermoneutral environment, and four in the heat, revealing no statistically significant differences across the three environments (*p* = 0.710). When considering all participants simultaneously, chi-square analysis showed that there were no differences in the number of participants who demonstrated CIVD reactions in the finger during exposure in the hot environment (n = 9) in comparison to the thermoneutral (n = 8; *p* < 0.001) and cool (n = 9; *p* < 0.001) environments. Overall, we found no differences in the number of participants demonstrating CIVDs in the fingers between sexes and across environments (*p* = 0.757).

### 3.2. Characteristics of CIVD

The characteristics of CIVD waves across groups and environments are shown in Appendix A (Table A1) as well as in Figure 4, and Appendix A Figure A1 and Figure A2. For the toes, as the male participants showed no CIVD in the cool or the thermoneutral environments, we were only able to compare the CIVD characteristics between groups for the CIVD waves observed in the hot environment. Thus, for the hot environment we found that the first toe demonstrated small effect sizes for the minimum (d = 0.35) and maximum (d = 0.39) temperatures as well as for the onset time (d = 0.39) between men and women (Table A1) while there were large effect sizes for the peak time (d = 0.94), the average temperature (d = 0.87), and the amplitude (d = 0.85) with women reaching faster peak time, higher average temperature, and lower amplitude. The third toe demonstrated medium effect sizes for the T_min_ (d = 0.90) and very small for T_max_ (d = 0.09), while there were no substantial effect sizes in the T_onset_ and T_peak_ (very small to small effect sizes; d = 0.26 to 0.10) between the two groups. Women demonstrated lower T_avg_ (large effect size; d = 0.81) and higher amplitude (medium effect size; d = 0.46) compared to men. Finally, for the fifth toe, T_onset_ was earlier for women compared to men, demonstrating a large effect size (d = 0.84). At the same time, women demonstrated higher (by ~2 °C) toe T_avg_ compared to their male counterparts (medium effect size; d = 0.66).

For the second finger, in the cool environment women demonstrated lower T_min_ (d = 1.74) and T_max_ (d = 0.56), delayed T_onset_ and T_peak_ (very small to small effect sizes; d = 0.02 to 0.40), as well as negligible differences in T_avg_ and the amplitude in comparison to men. In the thermoneutral environment, women demonstrated lower T_min_, T_max_, T_avg_, and amplitude (d = 0.82 to 1.43), as well as earlier T_onset_ and T_peak_ (medium to huge effect sizes; d = 0.51 to 2.94) compared to men. In the hot environment, women demonstrated higher T_min_, T_max_, T_avg_, and amplitude (medium to very large effect sizes; d = 0.65 to 1.74) as well as earlier T_onset_ (d = 0.78) compared to men, while T_peak_ was similar between sexes (d = 0.06).

### 3.3. Cardiovascular Strain

Women demonstrated higher cardiovascular strain during the entire protocol across all environments in comparison to their male counterparts. Specifically, women presented higher HR values (medium to large effect sizes; d = 0.58 to 1.81; Table A2, Figure A3, Figure A4 and Figure A5) in comparison to men. Moreover, they also demonstrated higher diastolic blood pressure (very small to large effect sizes; d = 0.11 to 0.99) and lower systolic blood pressure (very small to large effect sizes; d = 0.05 to 0.73) compared to men. In the cool environment, during the first minute of cold-water immersion, women experienced higher cardiovascular strain (as expressed by delta differences from the last minute of warm-water immersion) in comparison to men (Figure 3).

### 3.4. Other Physiological Responses

The physiological parameters monitored during the protocol are shown in Appendix A (Table A2; Figure A3, Figure A4 and Figure A5). In most cases, the differences observed between groups did not reach the Bonferroni adjusted threshold of *p* < 0.008 for post hoc comparisons, yet a number of moderate or large effect sizes were detected. Specifically, the T_gi_ was lower for men compared to women in the cool environment during baseline (d = 0.86), in the thermoneutral environment during cold-water immersion (d = 0.88) and recovery (d = 0.73), as well as in the hot environment during baseline (d = 1.40), warm-water immersion (d = 1.01), and cold-water immersion (d = 0.76).

No substantial differences were identified between the T_sk_ of men and women (very small to medium effect sizes; d = 0.01 to 0.68), with the female group demonstrating a slightly increased T_sk_ (Table A2). Nevertheless, it is important to note that across all environments and phases, women had higher chest skin temperature (large to very large effect sizes; d = 0.80 to 2.11) and lower limb (gastrocnemius) temperature (small to medium effect sizes; d = 0.01 to 0.65) compared to men (Table A2). On the other hand, in most cases, similar T_t_ and SkBF in the immersed foot were observed between men and women across all environments (d = 0.02 to 0.69; Table A2).

No marked and/or meaningful differences were observed between men and women in the sweat rate of the forehead and the leg (gastrocnemius), except for the thermoneutral environment where women showed markedly higher forehead sweat rate (medium to huge effect sizes, d = 0.66 to 2.47) and lower leg sweat rate for the hot environment during recovery (d = 0.91).

### 3.5. Perceptual Data

Pain and distress were similar between men and women across all environments (very small to medium effect sizes; d = 0.01 to 0.59; Table A2). Compared to men, the tactile sensitivity of women tended to be slightly decreased when assessed using the Semmes–Weinstein monofilaments (very small to medium effect sizes; d = 0.01 to 0.69) and slightly increased when assessed using a digital esthesiometer (very small to large effect sizes; d = 0.01 to 0.80; Table A2). This difference suggests that adoption of both methods provides a more robust assessment of tactile sensitivity in relation to CIVD. During the cold-water immersion and the recovery, women reported being more thermally uncomfortable compared to men in the cool and the thermoneutral environments (small to large effect sizes; d = 0.13 to 0.93), and this was reversed in the heat (no to medium effect sizes; d = 0.00 to 0.64). Finally, in terms of thermal sensation, men reported feeling slightly warmer than women in the cool environment (small to large effect sizes; d = 0.17 to 0.98), and these results were reversed in the hot environment (small to medium effect sizes; d = 0.14 to 0.58; Table A2).

## 4. Discussion

For a matched thermal stress, women in the present study experienced an elevated cardiovascular strain and higher frequency of CIVD reactions in comparison to their male counterparts. Specifically, the female group had higher HR and twice as many CIVD reactions compared to their matched male counterparts during exposure across all environments, while they had increased mean arterial pressure in the cool environment and almost the same mean arterial pressure in the other two environments (i.e., thermoneutral and hot). When both fingers and toes were considered, we found no statistically significant effects of sex in the overall CIVD occurrence. However, females demonstrated a statistically higher occurrence of CIVD was in the toes across all environments accompanied by an anticipated increase in the local SkBF of the immersed foot, which may be triggered as a response to the lower T_t_.

The few previous studies [10,26,27,28] evaluating sex differences in the occurrence of CIVD indicate no systematic difference between women and men. However, these studies investigated the occurrence of CIVD responses only in the fingers of their participants, and their results are in line with our findings of no substantial sexual differences in CIVD reactions in the fingers. In contrast, we show for the first time that women experienced twice as many CIVD reactions compared to men in the toes. This may be explained through two theoretical frameworks. First, previous studies state that there are potential differences in vasomotion between men and women due to hormonal status [49] or due to an increase in sympathetic outflow to the cutaneous circulation in women [23]. Furthermore, other studies indicated that there are no sex differences in cutaneous vasodilation [50], and that the sex differences are probably due to epidermis thickness [51] or the size of the hand and foot rather than due to the hormonal status [52]. Thus, the second, and probably more important, theory is that these differences may be a result of the smaller surface area which characterizes the immersed feet of women [52]. As a rule, the average woman (531 cm^2^) has approximately an 18% smaller foot compared to the average man (650 cm^2^) [53], which means that female feet may cool more quickly when exposed to cold stimuli, a notion that was previously proposed by others [52]. This may, therefore, affect local mechanisms that have been reported to induce CIVD [8], resulting in more frequent CIVD reactions.

Modern workplace practices involve females in the same work tasks as men, and female workers can be currently found doing jobs requiring frequent immersion of their limbs to cold water, such as butchering [30], fishing [31], and performing army-related tasks [32,33]. In the coming decades, more women are expected to enter the world of male-dominated work; therefore, more studies should be directed toward identifying workplace strategies for achieving gender equity. In this light, the literature suggests that sex should be considered an important contributing factor capable of modifying the physiological heat strain experienced by workers [54,55,56]. Yet, the majority of relevant studies investigate the impact of occupational heat-stress, not cold exposure. Only a handful of studies on army-related tasks or occupations such as fishing or butchering have assessed the risk for cold injuries in the extremities during exposure to cold [57,58,59]. Some of these studies found that women have increased risk for cold injuries compared to men [60,61] but none of them mention protective footwear for occupations such as fishing, butchering, and especially the disparate cold injury risks for males and females. These sex differences were further confirmed in the present study, indicating that foot clothing and shoe industries should contemplate implementing new technologies that will consider sex differences in an attempt to better protect worker health and well-being. In addition to this, during exposure to cool environments, female workers may be placed in jobs that do not require frequent exposure of their limbs to cold water. It is important to note that, although this is the first study to examine sex differences in cardiovascular stress and CIVD responses in feet and hands during cold-water immersion, it involved monitoring a small group of Caucasian individuals that were not habituated to frequent exposure of their limbs to cold-water immersion. Future studies should consider investigating how inter- and intra- individual factors such as age, ethnicity, and body composition may contribute to the occurrence of the CIVD responses in the feet. Finally, it is worth noting that the lack of heterogeneity and sex differences in CIVD reactions in the hands is likely due to the single measurement site, contrary to the toes where we used three sites and reported significant sex differences. Nevertheless, it is important to note that all previous studies on the topic reported no effects of sex on CIVD in the fingers [10,26,27,28].

## 5. Conclusions

The present investigation demonstrated that the studied women experienced elevated cardiovascular strain and higher frequency of CIVD reactions, particularly in the toes, compared to their male counterparts, especially when exposed to cool environments.

## Figures and Tables

**Figure 1 biology-11-01054-f001:**
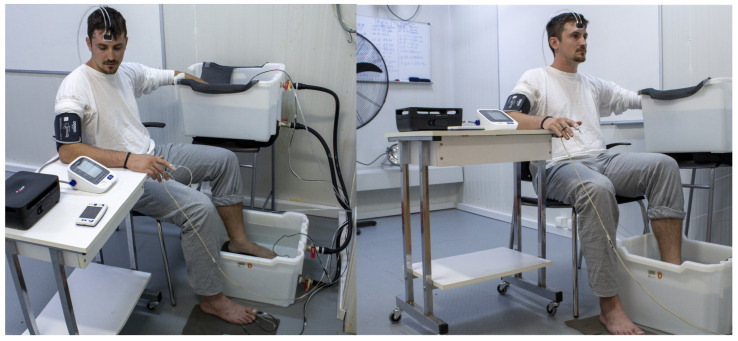
A male participant during the data collection. Note: the right sleeve of the shirt was cut at elbow height to avoid constricting blood flow. The blood pressure cuff was comfortably placed at the upper arm without constricting blood flow except when assessing arterial blood pressure. Photograph used with permission from the participant featured.

**Figure 2 biology-11-01054-f002:**
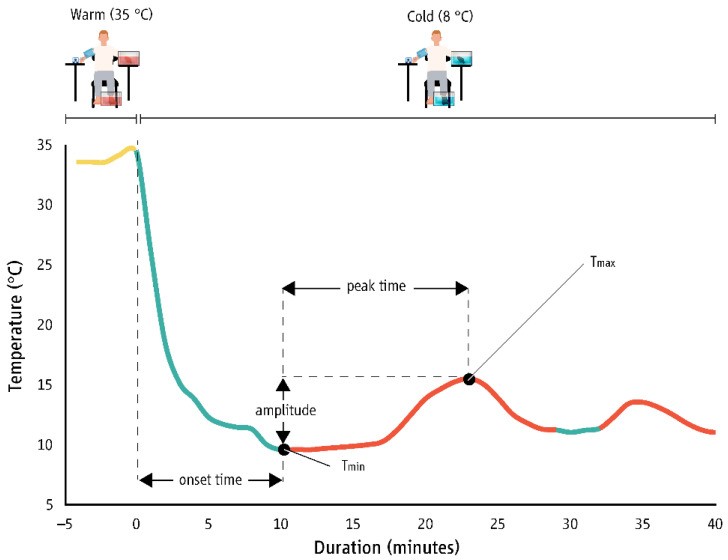
Representative data for two cold-induced vasodilation (CIVD) reactions (red color) in the index finger of a male participant. Onset time (green color) corresponds to the time needed to reach minimum temperature (T_min_) prior to the onset of CIVD. Amplitude represents the difference between T_min_ and the highest temperature during the CIVD (T_max_). Peak time corresponds to the time interval between T_min_ and T_max_.

**Figure 3 biology-11-01054-f003:**
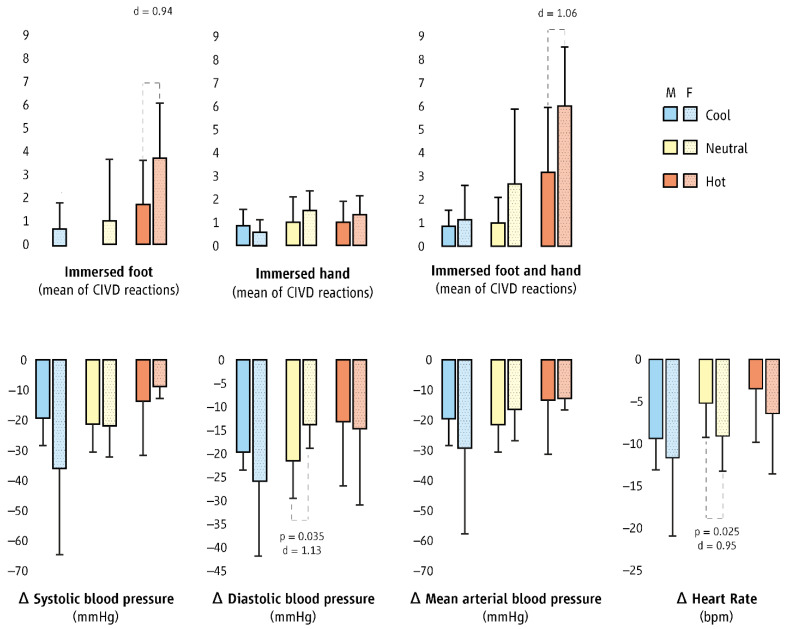
Differences (mean ± SD) between men (M) and women (F) in the occurrence of cold-induced vasodilation reactions (CIVD; top bars) and cardiovascular strain during the first minute of cold-water immersion (bottom bars) across cool (blue bars), thermoneutral (yellow bars), and hot (orange bars) environments. Cardiovascular strain during the first minute of cold-water immersion was expressed as the delta difference from the last minute of warm-water immersion and the first minute of cold-water immersion.

**Figure 4 biology-11-01054-f004:**
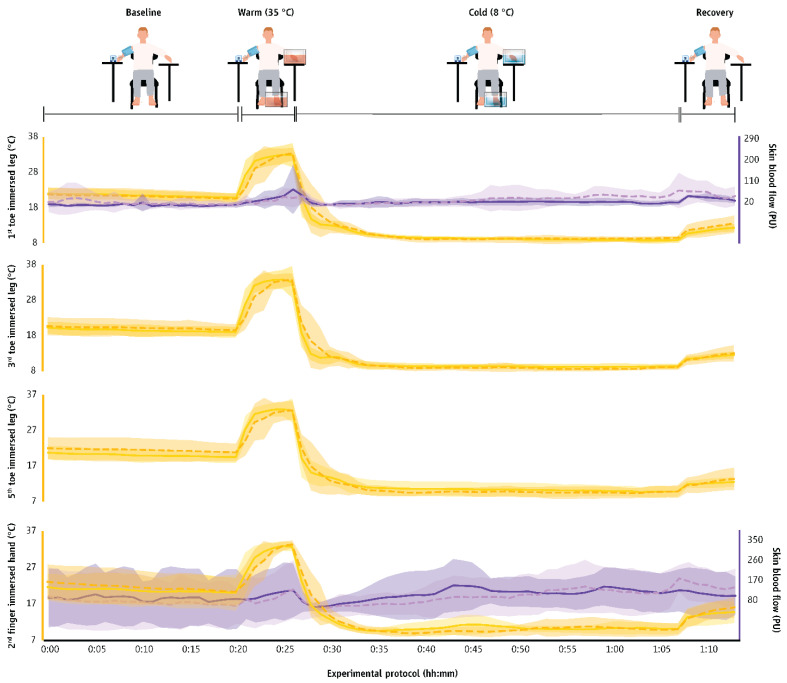
Skin temperature and blood flow (mean ± SD) in the fingers and toes during exposure to the cool environment (16 ± 1 °C) in the two groups. The first 20 min (00:00 to 00:20) indicate data collected during the baseline phase, the next five min (00:20 to 00:25) indicate responses during the warm-water immersion, the next 40 min (00:30 to 00:70) indicate responses during the cold-water immersion, and the final five min (00:70 to 00:75) show responses during the recovery phase. Finger and toe temperatures are indicated with continuous yellow lines in men and with dashed yellow lines in women. Finger and toe skin blood flow data are indicated with continuous purple lines in men and with dashed purple lines in women.

**Table 1 biology-11-01054-t001:** Anthropometric characteristics (mean ± SD) of the participants in the two groups.

Group	n	Age (Years)	Weight (kg)	Height (m)	BMI (kg/m^2^)	BSA (m^2^)
Men	7	34.0 ± 12.2	81.1 ± 17.7	1.76 ± 0.5	26.0 ± 5.0	2.0 ± 0.2
Women	7	32.1 ± 8.5	75.5 ± 15.9	1.72 ± 0.5	25.4 ± 5.1	1.9 ± 1.8

Note: BMI: body mass index; BSA: body surface area.

**Table 2 biology-11-01054-t002:** Heat map presenting the number of participants that demonstrated a CIVD reaction for each minute of the cold immersion during exposure in the cool environment (16 ± 1 °C). Colors indicate frequencies. Gray color indicates lack of CIVD.

		Time (min)																																								
		0	1	2	3	4	5	6	7	8	9	10	11	12	13	14	15	16	17	18	19	20	21	22	23	24	25	26	27	28	29	30	31	32	33	34	35	36	37	38	39	40
Women	Second finger											1	1	2	2	2	3	4	4	4	4	4	4	4	4	3	3	3	3	3	2	2	1	1	1	1	1	1	1	1	1	
	First toe														1	1	1	1	1	1	1	1	1	1	1	1	1	1	1	1	1	1	1	1	1	1	1	1	1			
	Third toe																					1	1	1	1	1																
	Fifth toe																				1	1	1	1	1	1																
	First toe *																																									
Men	Second finger									2	2	2	2	3	4	4	4	4	4	5	5	5	5	5	5	5	3	3	3	3	3	3	3	2	2	2	2	2	2	2	2	2
	First toe																																									
	Third toe																																									
	Fifth toe																																									
	First toe *																																									

Notes: * = non-immersed foot (all other fingers and toes refer to the immersed limbs).

**Table 3 biology-11-01054-t003:** Heat map presenting the number of participants that demonstrated a CIVD reaction for each minute of the cold immersion during exposure in the thermoneutral environment (23 ± 1 °C). Colors indicate frequencies. Gray color indicates lack of CIVD.

		Time (min)																																								
		0	1	2	3	4	5	6	7	8	9	10	11	12	13	14	15	16	17	18	19	20	21	22	23	24	25	26	27	28	29	30	31	32	33	34	35	36	37	38	39	40
Women	2nd finger									1	1	2	4	5	5	5	5	5	5	4	3	1	1	2	2	2	2	2	2	4	4	4	3	3	3	3	2	2	2	1	1	1
	1st toe										1	1	1	1	1	1	1	1	1	1																			1	1	1	1
	3rd toe											1	1	1	1	1																		1	1	1	1	1	1	1	1	
	5th toe											1	1	1	1	1	1	1	1	1	1				1	1	1	1	1	1	1	1	1	1	1	1	1	1	1	1	1	1
	1st toe *																																									
Men	2nd finger							1	1	1	2	2	3	3	3	3	3	3	3	3	3	3	3	3	3	3	3	3	2	2	3	2	2	2	3	3	3	3	3	2		
	1st toe																																									
	3rd toe																																									
	5th toe																																									
	1st toe *																																									

Notes: * = non-immersed foot (all other fingers and toes refer to the immersed limbs).

**Table 4 biology-11-01054-t004:** Heat map presenting the number of participants that demonstrated a CIVD reaction for each minute of the cold immersion during exposure in the hot environment (34 ± 1 °C). Colors indicate frequencies. Gray color indicates lack of CIVD.

		Time (min)																																								
		0	1	2	3	4	5	6	7	8	9	10	11	12	13	14	15	16	17	18	19	20	21	22	23	24	25	26	27	28	29	30	31	32	33	34	35	36	37	38	39	40
Women	2nd finger								1	1	1	3	4	4	5	5	5	5	5	5	5	5	5	5	5	5	5	5	5	5	5	5	5	5	3	3	3	3	3	2	2	2
	1st toe									1	2	2	2	2	2	3	4	4	4	4	3	3	4	4	4	3	4	2	3	3	3	3	4	4	4	4	4	3	3	3	2	2
	3rd toe							1	1	3	3	3	4	4	5	5	5	5	4	4	4	3	2	3	2	3	3	3	3	4	4	4	4	4	3	3	3	2	1	1	1	1
	5th toe									2	3	3	4	5	5	5	5	5	5	4	4	4	3	2	2	2	2	2	2	2	2	2	3	2	2	2	2	1	1	1	1	
	1st toe *						1	1	1	1	1	1	1	1	1	1	1	1	1	1	1	1			1	1	1	1	1	1												
Men	2nd finger							1	1	1	1	1	1	2	2	3	3	3	3	2	2	3	3	3	3	3	3	3	4	4	4	4	4	4	4	4	4	3	2	2	2	2
	1st toe													2	2	2	2	2	2	2	2	2	2	2	1	2	2	2	2	2	2	2	1	1	1	1	1	1	1	1		
	3rd toe													1	2	3	3	3	3	2	2	2	3	3	2	2	2	1	1	1	1	1	1	1	1	1	1	1	1	1	1	1
	5th toe												1	2	2	2	2	2	2	2	2	2	2	3	1	1	1	1	1	2	2	2	2	2	1	1	1	1	1			
	1st toe *				1	2	2	2	2	2	1	1	1	1	1	1	1	1	1	1	2	1	1	1	1	1	1	1	1	1	1	1	1									

Notes: * = non-immersed foot (all other fingers and toes refer to the immersed limbs).

## Data Availability

The data that support the findings of this study are available from the corresponding author, upon reasonable request.

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
