# Peer review of "Cardiovascular Stress and Characteristics of Cold-Induced Vasodilation in Women and Men during Cold-Water Immersion: A Randomized Control Study"

_biology, 2022, doi:10.3390/biology11071054_

Round 1

Reviewer 1 Report

A thoroughly-researched and well thought out study. Data will add to the literature. Colour figures 3 and 4 are particularly striking, and informative.

Author Response

Dear Reviewer, thank you very much for your supportive comments and your time to review our paper.

Please find attached below our responses to your comments.

Reviewer 2 Report

This is a well written study that examines the role of sex on CIVD appearance on cool, thermoneutral and warm environment. The authors support the significant role of gender on CIVD frequency especially in lower leg. Some suggestions / observations are following.

·      My main concern related to the number of sites measured in hand and foot, given the high heterogeneity between hand and foot and across fingers. In the hand there was only one site, and this is probably the reason for the conclusion “we found no differences in the number of participants demonstrating CIVDs in the fingers between sexes and across environments (p = 0.757) (lines 298-300). Maybe this should be referred as a limitation of the study.

·      Was there any circulation of the water in the tanks? This is because the responses of the fingers in the steering water are of course different and probably affects CIVD. If not, this should be reported as possible limitation.

·      In the results is reported that the heat strain was similar. I wander if you can support this because the CV strain was different, and this could also affect temperature regulation.

·      Fig. 3: is there any meaning of the merged number of CIVD reactions? 

·      Fig 2: the indication of Tmin to my opinion is in a different place.

·      Table 1: maybe you should report the immersed area of hand and foot, since you support the role of this factor for the different responses across sexes. 

·      In the discussion you report that “previous studies state that there are potential differences in vasomotion between men and women due to hormonal status” and you suspect that it is one of the reasons for the observed differences. However, in the method section you report no control of the menstrual cycle since not seem to affect CIVD. To me, this a contradictory issue that needs to be resolved. Also, maybe this is an explanation for the higher SwR in the forehead found in women compared to man.

Author Response

(The authors gave the same response as above.)

Reviewer 3 Report

Dear authors,

congratulations for your work. You tried to show any possible differences between genders in terms of cardiovascular stress and vasodilation responses to cold water immersion.

In general the article is well written and structures but in my oppinion your study is a study-case due to the small sample, and you have to considered when you made your conclusions.

Here are my suggestions:

1) Add study-case in your title

2) Line 49-52 Add references to support this setence.

3) Line 64 What is your goal? Based on your sentence why this is important to have answers? I do not see why study this.

4) Add you study design in a specific section in methods. It will be more clear if any other researcher want to apply your protocol.

5)Which clothes were permited? how did you assume that this do not influenciated the response of the participants?

6) Line 422-424 - In my oppinion you can not state this because your study is a study case because of your small sample.

Author Response

(The authors gave the same response as above.)

Reviewer 4 Report

Thank you for submitting your valuable manuscript to this journal.

The aim of the present study was to compare CIVD reactions between a group of matched men and women in different environmental conditions. This is a high standard original study with high-tech instruments and measurements.

There are some points that may improve the study report for publication:

1-      It would be great if the authors could further explain more about the impact of the CIVD response in humans and its applications in occupational medicine and career advice.

2-      On materials and methods, it is better to firstly describe the participants’ inclusion and exclusion criteria, then on statistical methods, you can explain the sample size calculations.

3-      The sample size calculation was based on Tf max in two different environmental conditions and the result was six participants per group. Since you have three conditions in this study, the required sample size would be 18 subjects. Moreover, the aim of the study is to compare men and women responses, so you need even more participants for this purpose.
It is recommended to mention this study limitation at the end of discussion.

4-      Line 322, is here the neutral condition or the thermoneutral environment? I realized that the thermoneutral environment was just for the baseline measurements and it is different from neutral condition which is one of the three experimental conditions! Refer to line 122 for thermoneutral environment characteristics which is different from neutral condition in line 109.
Are the baseline temperatures in thermoneutral environment (apart from the neutral condition) reported in this study?  Did the authors calculate the difference of each measurement with its baseline value in thermoneutral environment?

Have the absolute values of Tf or Tt in each condition been used for the analyses, or correctly their differences from thermoneutral environment were considered for this purpose? It is true that there is not a significant difference in baseline values, but generally, using the difference values rather than the absolute values can make the results more reliable.

5-      At the beginning of discussion, it must be noted that there were no statistically significant differences between sexes in the overall CIVD occurrence. The difference was significant only in Tt. Line 266.

6-      The conclusion of the study needs to be revised according to the non-significant findings. You can just mention that there might be a trend towards the ...

Some minor points:

1-      Line 126, it is better to mention “entered their randomly allocated environmental chamber”

2-      Line 136, the sum of tests duration is 70 minutes!

3-      Line 287, no “toes” CIVD reactions

Author Response

(The authors gave the same response as above.)

Round 2

Reviewer 3 Report

Dear authors,

you have attended some of my suggestions and the article improved since the last version. Although, the G*power should be used for sample size calculation. I suggest the authors to perform this procedure.

Author Response

Dear Reviewer,

Thank you for your efforts in reviewing the manuscript and for your constructive and helpful comments. 

Please find our response in the attachment.
